# Imaging the dynamic influence of functional groups on metal-organic frameworks

Boyang Liu [1,2], Xiao Chen [1,3] ✉, Ning Huang[1,4], Shaoxiong Liu[1], Yu Wang [1], Xiaocheng Lan[1], Fei Wei[1] & Tiefeng Wang [1] ✉

Metal-organic frameworks (MOFs) with different functional groups have wide applications, while the understanding of functionalization influences remains insufficient. Previous researches focused on the static changes in electronic structure or chemical environment, while it is unclear in the aspect of dynamic influence, especially in the direct imaging of dynamic changes after functionalization. Here we use integrated differential phase contrast scanning transmission electron microscopy (iDPC-STEM) to directly 'see' the rotation properties of benzene rings in the linkers of UiO-66, and observe the high correlation between local rigidity and the functional groups on the organic linkers. The rigidity is then correlated to the macroscopic properties of $CO_2$ uptake, indicating that functionalization can change the capability through not only static electronic effects, but also dynamic rotation properties. To the best of our knowledge this is the first example of a technique to directly image the rotation properties of linkers in MOFs, which provides an approach to study the local flexibility and paves the way for potential applications in capturing, separation and molecular machine.

Metal-organic frameworks (MOFs) are formulated by metal clusters and organic linkers with periodic frame structure, and have been applied in catalysis[1,2], separation[3–7] and nanodevices[8,9]. The functionalization of organic linkers significantly expands the variety of MOFs, and brings unique properties[10,11]. UiO-66 is one of the most stable MOFs[12], and has applications in $CO_2$ capture[13], photocatalysis[14] and sensing[15]. The addition of functional groups to the p-benzenedicarboxylic acid (BDC) linkers creates a large class of UiO-66-X materials with various properties[16,17]. For instance, amino modified UiO-66-$NH_2$ is widely used in the synthesis of single atom catalyst (SACs), due to the strong interaction between amino groups and metals[18–20]. Other functionalized UiO-66-X can also be applied in pharmaceutical fields like drug delivery[21,22].

However, the understanding of how functional groups influence the MOFs remains insufficient. Previous studies mainly focused on the static changes in electronic structure or chemical environment, while it

is unclear regarding the influence on the dynamic properties, which is essential for molecular machines[23] and gas separations[5]. Lillerud et al. [17] reported that thermal and chemical stability of UiO-66-X remains unchanged after functionalization. Walton et al. [16] attributed the different adsorption properties of UiO-66-X to the combination of polarity and small functional group size. The different catalytic performance of UiO-66-X is usually ascribed to the unique electronic properties of functional groups[24,25]. The influences of functionalization on the dynamic properties of UiO-66-X have been rarely studied, although such influences can also change the adsorption or catalytic properties.

The dynamic properties of MOFs are widely known as the flexibility, and has been studied for their unique breathing and swelling properties[26,27]. The rotation of benzene rings in BDC linkers, which is also called π-flipping, is also an important component of the flexibility of MOFs, and has been studied for gas separation[5] and molecular

[1]Beijing Key Laboratory of Green Chemical Reaction Engineering and Technology, Department of Chemical Engineering, Tsinghua University, Beijing 100084, PR China. [2]School of Chemistry and Chemical Engineering, Nanjing University of Science and Technology, Nanjing, Jiangsu 210094, China. [3]Ordos Laboratory, Ordos, Inner Mongolia 017000, China. [4]Sinopec Economics and Development Research Institute Company Limited, Beijing 100029, China. ✉e-mail: chenx123@tsinghua.edu.cn; wangtf@tsinghua.edu.cn

rotors[28]. [2]H-NMR (nuclear magnetic resonance) is one of the limited ways to understand the π-flipping process[29], and has been applied in UiO-66[30], MOF-5[31], IRMOF-2[32] and MIL-53[33]. However, the spectroscopy provides only indirect and statistical information of the MOF samples. The integrated differential phase contrast scanning transmission electron microscopy (iDPC-STEM) technology has been applied in characterization of zeolites[34–36] and MOFs[37–39], with the advantages of simultaneous imaging of light and heavy elements. With iDPC-STEM, researchers are able to directly 'see' the organic linkers bridging the metal nodes in MOFs in real space. However, the imaging of MOFs still remains challenging because they are very sensitive to the electron beams, and are unstable even under an ultra-low dose[40]. The identification of metal nodes is relatively easy[39], while the atomic imaging of organic linkers is particularly difficult. Since the atomic identification of organic linkers using high-resolution transmission electron microscopy (HRTEM) in 2018[41], few literatures have successfully characterized the linkers in MOFs atomically[42], and basically no literature has studied the dynamic properties of linkers using electron microscopy.

The benzene rings of BDC linkers in UiO-66 are considered rotatable at room temperature (RT)[43,44], while the rotation properties with different functional groups have been rarely studied. In this work, we studied the π-flipping properties of UiO-66-X with different functional groups on BDC-X linkers. Through iDPC-STEM imaging, we observed that functional groups are highly related to the local rigidity of BDC linkers, and might further influence the macroscopic properties. With suitable functional groups, some UiO-66-X samples show local rigidity that the benzene rings of BDC-X linkers show basically identical orientations, which, to the best of our knowledge, has not been reported.

## Results

### Synthesis and structure characterization of UiO-66-X

UiO-66 are formed by Zr nodes and BDC linkers, with tetrahedron and octahedron pores (Fig. 1a). Viewed from the direction of [110], UiO-66 will be projected into a rhombus tessellated pattern. The four vertices of

a cell are all Zr nodes, and the four edges are BDC linkers with a degree of 30°. The short diagonal is another column of BDC linkers, with their benzene rings parallel to the [110] projections. The short diagonal passing through two carboxyl C and parallel to the (110) facet is denoted as the short axis (green arrow in Fig. 1a), while the long diagonal perpendicular to the benzene rings of the BDC linkers is denoted as the long axis (blue arrow in Fig. 1a). If the benzene rings of BDC linkers do not rotate around the $C_2$ axis (Supplementary Fig. 1), the intensity profile along the long axis will contain one narrow peak in the middle, which corresponds to the benzene rings degenerating into a line along the short axis. However, if the benzene ring flips with a degree, the narrow line will be blurred and broadened to a larger bright spot.

When changing the linkers to other BDC-X linkers with different functional groups, we can obtain UiO-66-X samples, which have similar structure to traditional UiO-66. To be consistent, we denote UiO-66 without any functional group as UiO-66-H, so as the BDC-H linkers. Through scanning electron microscopy (SEM), we found that the addition of functional groups rarely changes the morphology that all UiO-66-X samples appear as regular octahedrons (Supplementary Fig. 2). X-ray powder diffraction (XRD) results also confirm that the UiO-66-X samples have the same crystal structure with *fm*-3*m* symmetry (Fig. 1b), indicating that the functional groups of BDC-X linkers rarely influence the crystal structure. High angle annular dark-field scanning transmission electron microscopy (HAADF-STEM) images show smooth diamond-like structure from [110] projection (Supplementary Fig. 3), which are typical images of UiO-66 samples. Through these characterizations, we confirm that UiO-66-X samples are synthesized with similar structure, and the only difference is the functional groups.

The energy dispersive X-ray spectroscopy (EDS) is applied to confirm the existence of functional groups. The UiO-66-H samples show only Zr, C and O signals (Supplementary Fig. 4), while N, Cl, Br signals are obviously detected in UiO-66-NH₂, UiO-66-Cl and UiO-66-Br, respectively (Figs. 2a, d and Supplementary Figs. 5, 6). The

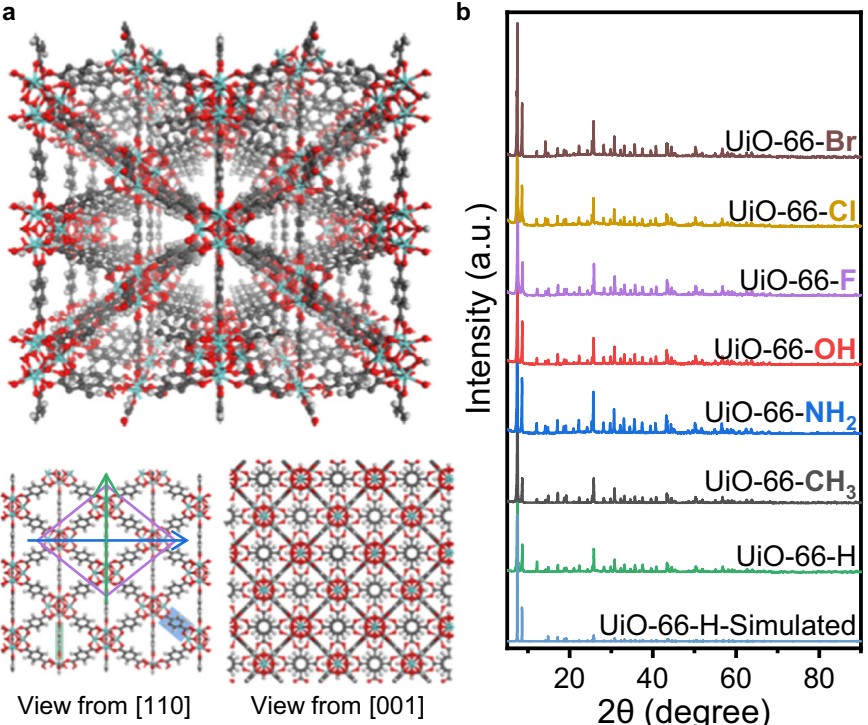

**Fig. 1 | Crystal structure of UiO-66-X. a** Schematic model of UiO-66. The views from [110] and [001] projections are shown below. The long and short axes from the [110] projection are marked by blue and green arrows, respectively. The grey, red and white atoms represent C, O and H elements, respectively. **b** XRD patterns of UiO-66-X and simulated results of UiO-66-H. Source data are provided as a Source Data file.

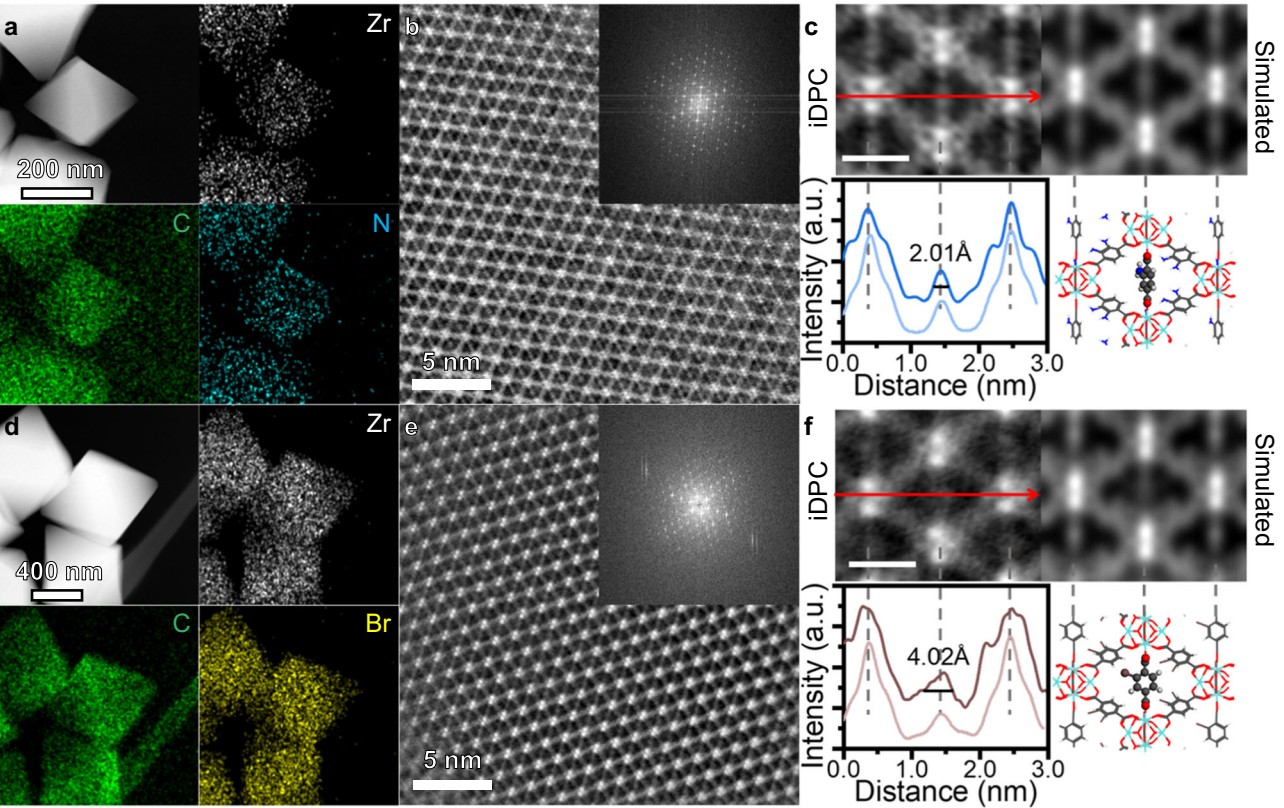

**Fig. 2 | Characterizations of UiO-66-NH₂ and UiO-66-Br.** HAADF-STEM images and corresponding EDS mapping of UiO-66-NH₂ (**a**) and UiO-66-Br (**d**). iDPC-STEM images of UiO-66-NH₂ (**b**) and UiO-66-Br (**e**) from [110] orientation. The FFT pattern is shown as inset. Magnified iDPC-STEM images and simulated results of UiO-66-NH₂ (**c**) and UiO-66-Br (**f**). The intensity profile along the long axis and the model of BDC-NH₂ rotation are shown below. The dark lines in the intensity profile are experimental results, while the light lines are the simulated results. The scale bar is 1 nm. The grey, red, white, blue and brown atoms represent C, O, H, N and Br elements, respectively. Source data are provided as a Source Data file.

composition of newly introduced elements is also consistent with the theoretical model, where all linkers were replaced by functionalized BDC-X linkers (Supplementary Table 1).

## The direct imaging of rotation properties of BDC-X linkers

Considering that HAADF-STEM images can hardly reflect the structure of organic linkers in UiO-66-X due to the e-beam damage, we applied the iDPC-STEM technology to image the BDC-X linkers for better understanding of the microscopic influence of functionalization. Taking UiO-66-NH₂ and UiO-66-Br as examples, we can simultaneous image the Zr nodes and BDC-X linkers from the [110] projection (Fig. 2b, e), where FFT patterns confirm the high resolution. The iDPC-STEM images are consistent with the schematic model that diamond-like cells are observed. For the UiO-66-NH₂ samples, the BDC-NH₂ linkers along the short axis are recognizable as four separate bright spots (Fig. 2c). These spots can be identified as two carboxyl groups and the upper and lower parts of benzene ring. The intensity profile along the long axis shows a peak with a full width at half maximum (FWHM) of 2.01 Å. The width of this peak represents the 'width' of the projection of BDC-NH₂ linkers in the (110) plane, which can be further correlated to the dynamic properties of UiO-66-X. For UiO-66-Br, however, the signals of BDC-Br are blurred from a narrow line to a larger bright spot (Fig. 2f). The asymmetric peak shape indicates that the functional groups of -Br may be biased in one direction. Moreover, the FWHM along the long axis increased to 4.02 Å. The broadening effect is attributed to the π-flipping of benzene rings in the BDC-Br linkers. Contrary to BDC-NH₂, the benzene rings of the BDC-Br linkers are nearly parallel to the (110) plane, leading to a large bright spot in the middle of both long and short axes. To the best of our knowledge

this is the first example of a technique to directly see the uniform and disordered orientations of benzene rings in UiO-66-NH₂ and UiO-66-Br, respectively, and the dramatic effect of functionalization on the dynamic properties is unexpected.

In addition to UiO-66-NH₂ and UiO-66-Br, we also use the iDPC-STEM technology to characterize the UiO-66-X (X = H, CH₃, OH, F, Cl) samples (Supplementary Fig. 7). The peak at the center of long axes shows varying degrees of broadening with different functional groups on the BDC-X linkers, and then indicates different dynamic properties of UiO-66-X (Fig. 3a–e). This broadening is ascribed to the different rotation properties of benzene rings, and the values of FWHM can be used as a quantitative description of the degree of benzene rotation. We summarized the intensity profiles of more than 20 areas, and calculated the average FWHM for different UiO-66-X samples (Fig. 3f, Supplementary Fig. 8). The experimental value is consistent with the simulated value, where a three-layered model is applied for simulation using the ToTEM software[45] (Supplementary Fig. 9). The rotation degrees of benzene ring for simulation are determined using 0.18 eV as the average rotation energy at RT, which will be discussed later. Through iDPC-STEM imaging, we can directly 'see' the different π-flipping properties of UiO-66-X. The rigidity of BDC-X linkers in UiO-66-X follows the trend of -OH > -NH₂ > -H > -CH₃ ~ -F > -Cl > -Br. This trend agrees well with the previously reported results of UiO-66-(OH)₂ > UiO-66-H > UiO-66-(CH₃)₂ obtained from ²H-NMR[30].

## Theoretical explanations of different dynamic properties of UiO-66-X

With the density functional theory (DFT), we calculated the rotation energies of BDC-X linkers and investigated the causes of

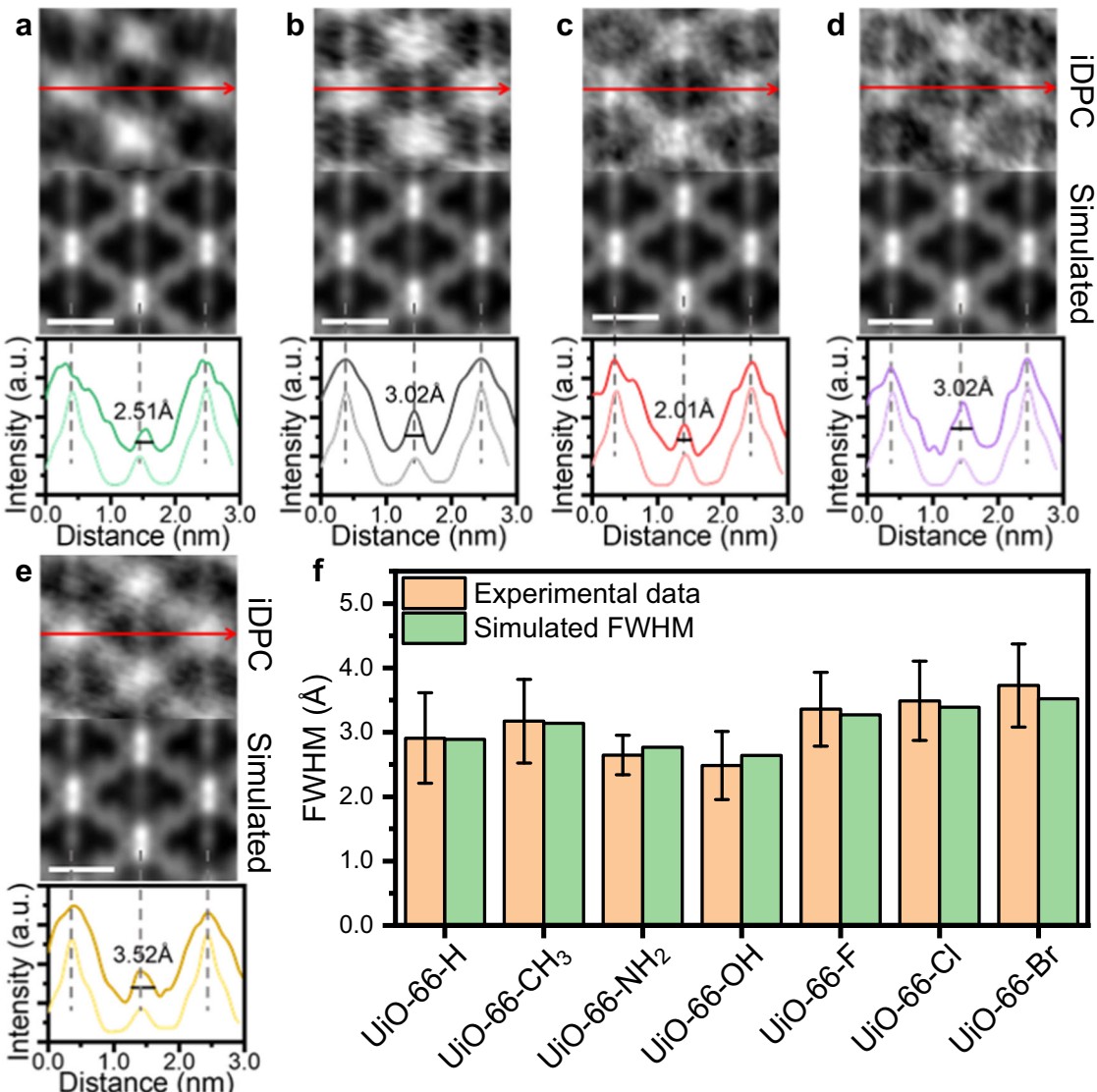

**Fig. 3 | Rotation properties of benzene ring in UiO-66-X.** iDPC-STEM images, simulated results and intensity profile along the long axis of UiO-66-H (**a**), UiO-66-CH₃ (**b**), UiO-66-OH (**c**), UiO-66-F (**d**) and UiO-66-Cl (**e**). The dark lines in the intensity profile are experimental results, while the light lines are the simulated results. The scale bar is 1 nm. **f** Summarized experimental and simulated FWHM of benzene rings along the long axis in UiO-66-X. The experimental FWHM is the averaged value obtained from more than 20 cells, and the error bar is the standard deviation. Source data are provided as a Source Data file.

different dynamic properties. The original model is built as a primitive cell of UiO-66-X, with the benzene ring perpendicular to the (110) plane. The total energy of this model is calculated as the energy reference. The benzene ring of the selected BDC-X linker is rotated with a certain degree around the $C_2$ axis, and the corresponding energy is calculated without ionic steps (Fig. 4a). The highest energy is always obtained when the benzene ring rotates by 90°, indicating that the state with benzene ring parallel to the (110) plane is the most unstable (Fig. 4c). This is obvious because two carboxyl groups and the benzene ring share a large π plane, which is destroyed by rotation and 90° of π-flipping represents the least conjugation. The calculated activation energy of π-flipping in UiO-66-H is 0.49 eV (47.4 kJ/mol), which is consistent with the experimental results in MOF-5 (47.3 kJ/mol)[31].

However, different functional groups show different influence on the π-flipping properties. The rotation energy at 90° can symbolize the difficulty of benzene rotation, where a higher energy represents more difficult π-flipping and more rigidity. Amino and hydroxyl groups show a stabilizing effect on the BDC-X linkers, while the other functional groups all enhance the flexibility (Fig. 4c). This is attributed to the

intramolecular hydrogen bond of BDC-NH₂ and BDC-OH (Supplementary Fig. 10). The distance between hydroxyl H and carboxyl O in BDC-OH is 1.64 Å, and the bond length between amino H and carboxyl O in BDC-NH₂ is 1.88 Å (Supplementary Table 2). On the contrary, the distance between methyl H and carboxyl O in BDC-CH₃ is much longer (2.37 Å), indicating the existence of hydrogen bonds in UiO-66-OH and UiO-66-NH₂, but not in UiO-66-CH₃. The density of states (DOS) and crystal orbital Hamilton population (COHP) results also verified the existence of hydrogen bonds (Supplementary Fig. 11). The positive value of −COHP represents the bonding effect, while the negative value of −COHP represents the antibonding effect[46]. The integrated value (−ICOHP) at the Fermi level represents the overall bonding condition. This value follows the trend of -OH...OOC > -NH₂...OOC > -CH₃...OOC, indicating that the strength of intramolecular hydrogen bond follows the same trend. This explains why UiO-66-OH is the most rigid sample, where the benzene can hardly rotate at RT. We build a virtual model of UiO-66-OH-Rev by changing the orientation of hydroxyl H in BDC-OH (Fig. 4b). The intramolecular hydrogen bond will be destroyed, and the rotation energy at 90° significantly decreases from 0.89 eV in UiO-66-OH to 0.29 eV in UiO-66-OH-Rev, indicating a higher flexibility (Fig. 4c).

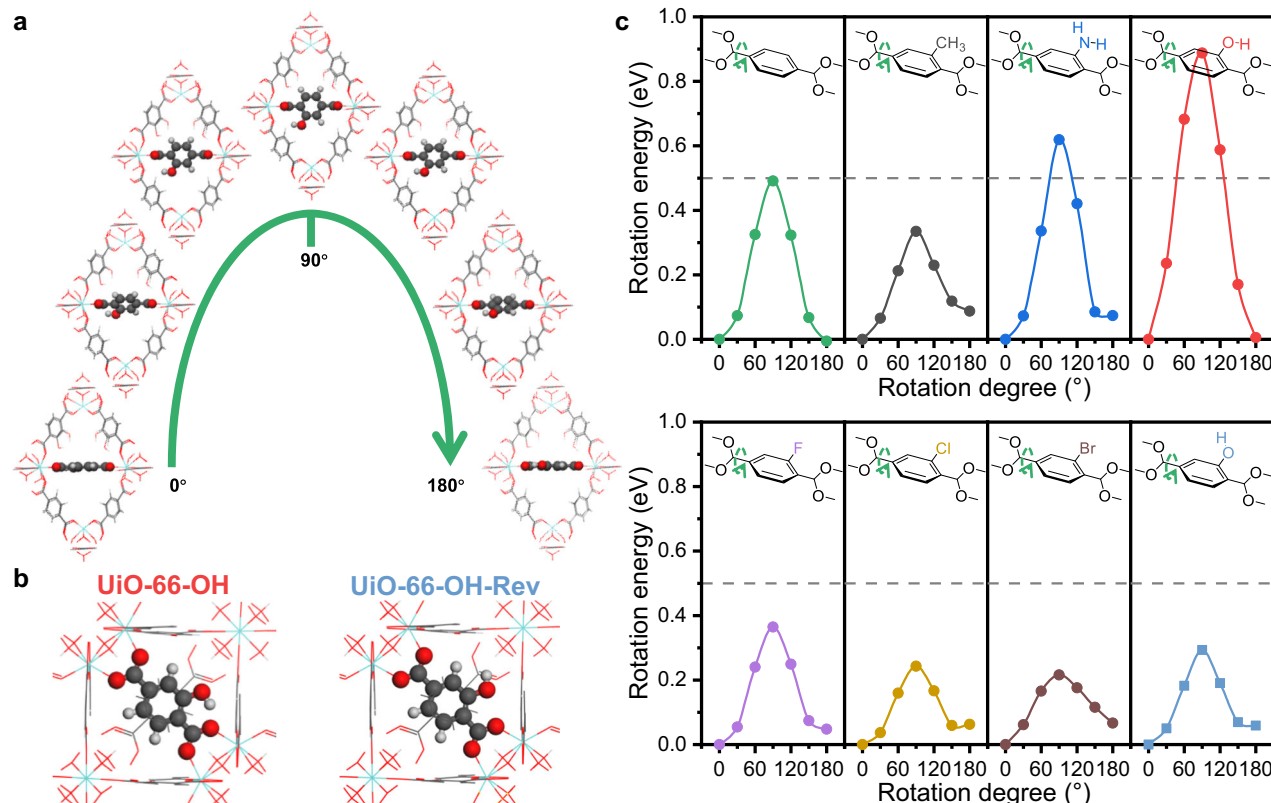

**Fig. 4 | Investigating the rotation of benzene ring in UiO-66-X via DFT calculations. a** Schematic model of benzene rotation in UiO-66-OH. **b** Schematic model of two types of UiO-66-OH, with different hydrogen direction in the hydroxyl group of BDC-OH linkers. **c** The calculated rotation energy of UiO-66-X against the rotation degree of benzene ring. The grey, red and white atoms represent C, O and H elements, respectively. Source data are provided as a Source Data file.

Therefore, the addition of functional groups, either electron donating (e.g. $-CH_3$) or electron withdrawing (e.g. $-F$, $-Br$), will destabilize the BDC-X linkers, due to the breaking of the symmetry of the conjugated structure. The benzene rings in BDC-X linkers will be more rotatable, and UiO-66-X samples show more flexibility. However, $-NH_2$ and $-OH$ groups are the exceptions, because they form intramolecular hydrogen bonds, which overcomes the destabilizing effects of functionalization and even inhibits the π-flipping. Thus, the UiO-66-$NH_2$ and UiO-66-OH samples show local rigidity at RT, and the BDC-X linkers become more identifiable under electron beam in iDPC-STEM images.

## Resolving the benzene rotation in UiO-66-X

The calculated rotation energy at 90° is also consistent with the FWHM obtained from iDPC-STEM. Higher rotation energy corresponds to smaller FWHM, indicating a more rigid BDC-X linker (Fig. 5a). We can also fit the experimental data using $FWHM = \frac{d}{2}\sin\left(\frac{E_0}{E_R} \times \frac{\pi}{2}\right) + \frac{d_{atom}}{2}$, where $d$ represents the width of BDC-X linkers and $E_R$ represents the rotation energy at 90°. If the rotation degree of benzene ring ($\theta$) is 0, the BDC-X linker along the short axis is parallel to the electron beam from [110] projection (Fig. 1a), as discussed at the beginning. The BDC-X linker is then projected to a line, but still with a width. We assume this broaden effect as a constant of $d_{atom}$. If the benzene ring rotates with a degree of $\theta$, the width of BDC-X linkers is projected to $d\sin\theta$. Then the projected bright spot of benzene ring should have a diameter of $d\sin\theta + d_{atom}$. The intensity profile along the long axis should contain a peak in the middle, with FWHM equals to $\frac{d}{2}\sin\theta + \frac{d_{atom}}{2}$ (Fig. 5b). We assume that the rotation of benzene ring originates from thermal motion, and the energy $E_0$ is basically constant at RT. The average rotation degree can be written as $\theta = \frac{\pi}{2}\frac{E_0}{E_R}$, where we further assume

that the rotation energy is basically proportional to the rotation degree.

Therefore, we can fit the relationship between FWHM obtained from iDPC-STEM experiments against the rotation energy at 90° calculated using DFT (Fig. 5a). The fitting parameters are $d$, $E_0$ and $d_{atom}$ (Supplementary Table 3). The fitted $E_0$ equals to 0.18 eV (-17 kJ/mol), which is a reasonable value of thermal motion energy at RT. For BDC-X (X ≠ H) linkers, the width is basically the same ranging from 3.59 to 4.08 Å, with an average value of 3.76 Å. The fitted value of $d$ is 3.78 Å, which is very close to the theoretical value. The consistency of experiments and calculations further confirm the reliability of our work.

Using the fitted parameter of $E_0$, we can calculate the average rotation degree of benzene rings in UiO-66-X (Supplementary Table 4). The simulated images of iDPC-STEM are then modeled using these rotation degrees. The simulated FWHM is consistent with the experimental data, indicating that the broaden of iDPC-STEM signals originates from the rotation of benzene rings (Fig. 3f).

## Correlating the local rigidity with the CO₂ capture

The replacement of functional groups in UiO-66-X can strongly affect the rotation properties and then local rigidity of these samples. The different macroscopic properties of the UiO-66-X samples with various functional groups are usually attributed to the static electronic effects[24,25], previously. However, the different dynamic properties can also significantly influence the macroscopic properties[47,48], the study of which is still very insufficient. Inspired by the fact that $CO_2$ adsorption can promote the flexibility of MOFs[49], here we briefly discuss the potential impacts of UiO-66-X on $CO_2$ capture, whose capability is highly related to their local rigidity.

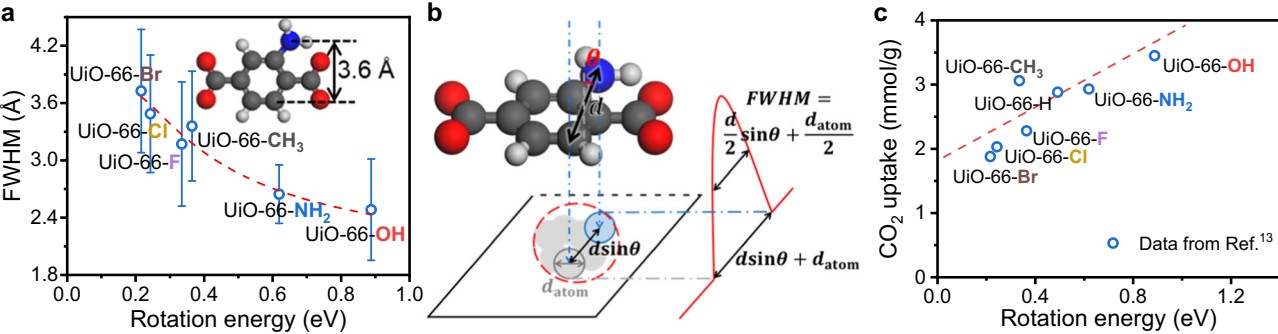

**Fig. 5 | Influence of rotation energies on the experimental FWHM and $CO_2$ adsorption properties. a** Fitted FWHM against rotation energies. The fitted data are measured with UiO-66-X (X ≠ H), and the correlation is

$$FWHM/\text{Å} = 1.89 \times \sin\left(\frac{0.18\,eV}{E_{Rotate}} \times \frac{\pi}{2}\right) + 1.82.$$ The corresponding $R^2$ is 0.96. The values of FWHM and $E_{Rotate}$ are obtained from iDPC-STEM images and DFT calculations,

while other parameters of $d$, $E_0$ and $d_{atom}$ are fitted. **b** Schematic model of the imaging of BDC-X linkers. **c** Correlation between $CO_2$ uptake and the rigidity of UiO-66-X. The values of $CO_2$ uptake are from Ref. [13] and the adsorption conditions are 1 bar and 0 °C. Source data are provided as a Source Data file.

Using the $CO_2$ uptake data from ref. [13], we can plot the relationship between $CO_2$ capture capability and the rotation properties of UiO-66-X (Fig. 5c). The difference in dynamic properties leads to different $CO_2$ capture performance, but their relationship has rarely been studied. Rigidity is considered beneficial to efficient collection of guest molecules, due to the larger number of identical porous units[48]. Our results agree with this view, and the most rigid UiO-66-OH sample has the highest $CO_2$ uptake amount. Although $CO_2$ is considered as acidic and has a quadrupole moment, the most basic UiO-66-$NH_2$ or the most polarizable UiO-66-Br did not show the highest $CO_2$ adsorption[50] (Supplementary Fig. 12). Instead, the more rigid of the UiO-66-X sample is, the more $CO_2$ it can adsorb generally. The $CO_2$ adsorption data from other researchers[51] are also consistent with this trend, although the absolute value of adsorption capacity is different (Supplementary Fig. 13). These results indicate that functionalization can influence the macroscopic properties, not only by tuning the electronic structure, but also by changing the dynamic properties. Moreover, when doping BDC-$NH_2$ to UiO-66-H, a higher fraction of BDC-$NH_2$ linkers leads to larger $CO_2$ uptake[52], which is also consistent with the trend that higher rigidity tend to benefit the adsorption of guest molecules. However, further investigation is required to better understand the mechanism of how local rigidity influences the $CO_2$ capture, and in situ inelastic- and quasi-elastic neutron scattering is one of the most accurate techniques[53].

In conclusion, we studied the dynamic influence of functional groups on UiO-66-X samples. Using iDPC-STEM technology, we are able to directly 'see' the rotation properties of benzene rings in BDC-X linkers, and the rigidity against π-flipping is highly related to the functional groups. To the best of our knowledge this is the first use of electron microscopy to image the rotation properties of organic linkers in MOFs, which is an important complement to the spectral method and provides an approach to understand the local flexibility of MOFs from a more direct perspective. Among the UiO-66-X samples, UiO-66-OH sample shows the highest local rigidity, which is attributed to the strong intramolecular hydrogen bond. The benzene rings in UiO-66-OH and UiO-66-$NH_2$ showed basically the same orientation at RT, which has not been reported. Moreover, the difference in the dynamic properties would be responsible for the different macroscopic properties, and we observe a positive relationship between $CO_2$ uptake and local rigidity of UiO-66-X. These results of various rotation properties of UiO-66-X pave the way for their potential applications in capturing of small molecules, separation of organic compounds and molecular machines.

## Methods

### Synthesis of UiO-66-X

The UiO-66-X with different functional groups are synthesized using basically the same solvothermal procedures, which is similar to the previously reported methods[20,54]. Typically, 0.135 mmol of BDC-X and equimolar $ZrCl_4$ are dissolved in 30 mL DMF containing 3.6 mL of HAc. The mixture is treated by ultrasound for 1 h, and then transferred to a 50 mL autoclave. The mixture is heated to 120 °C for 24 h. After cooling to RT, the products are collected through centrifugation after washing by methanol and DMF (1/4 v/v) for three times. The UiO-66-X samples are then dried under vacuum at 80 °C, and stored for later characterizations.

### The iDPC-STEM imaging

The iDPC-STEM images are taken on an FEI Titan Cubed Themis G2 300 microscope at 300 kV. The aberration is corrected using the following coefficients: C1 = −2.57 nm; A1 = 4.58 nm; A2 = 26.5 nm; B2 = 8.35 nm; C3 = 43.4 nm; A3 = 317 nm; S3 = 23.7 nm; A4 = 1.00 μm, D4 = 6.54 μm, B4 = 4.66 μm, C5 = 201 μm, A5 = 227 μm, S5 = 19.4 μm, and R5 = 51.0 μm. The convergence semi-angle is 15 mrad, and the collection angle is 4-22 mrad. The pixel size is 0.5027×0.5027 Å². The obtained images are applied with Radial Wiener filter to denoise and smooth.

### iDPC-STEM simulation

The image simulations are conducted using ToTEM software based on the multi-slice method[45]. Considering that we select the thin area of UiO-66-X for imaging, the simulation is modeled using three layers of cells. The average rotation degree $θ$ is calculated using 0.18 eV as the thermal motion energy. The BDC-X linkers in three layers are rotated by $θ$, 0° and $−θ$ respectively. The parameters for the image simulations are the same as those in the experiments.

### Other characterizations

The XRD patterns of UiO-66-X are collected by a Bruker D8 Advance powder X-ray diffractometer equipped with a Cu Kα radiation source in the 2$θ$ range of 5–90° at a ramping rate of 2°/min. The simulated patterns of UiO-66-H are obtained using the model from ref. [55]. The SEM images are obtained on JSM 7401 microscope with an electron emission voltage of 3 kV. The HAADF-STEM images and corresponding EDS elementary mapping are collected on FEI Titan Cubed Themis G2 300 microscope.

### DFT calculations

The DFT calculations are performed using Vienna ab initio simulation package (VASP)[56], with the Perdew-Burke-Ernzerhof (PBE) function[57].

The cutoff energy is set at 400 eV and the k-point grid is set at $1 \times 1 \times 1$ for structure optimization. The UiO-66-H is modeled based on the previous reports[55], and has been relaxed before further calculations. Other UiO-66-X models are built by replacing all the linkers from BDC-H to BDC-X. One primitive cell contains 6 Zr atoms and 6 BDC-X linkers. The rotation energy is calculated by rotating one of these 6 BDC-X linkers without moving any other atoms. The DOS is calculated using 2001 points. COHP analysis[46,58] is conducted using LOBSTER to better understand the bonding effects between H in functional groups and O in carboxyl groups. The calculated energy range is between −10 and 6 eV. A positive value of −COHP represents the bonding effect, while a negative value represents antibonding.

## Data availability
The authors declare that all relevant data supporting the findings of this study are available within the paper and its Supplementary Information files. Additional data are available from the corresponding authors upon request. Source data are provided with this paper.

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

## Acknowledgements

We thank Fang Lin and Yuming Ge from South China Agricultural University for their assistance in TEM simulation with ToTEM software. This work is also supported by Tsinghua National Laboratory for Information Science and Technology. Financial support from the National Natural Science Foundation of China (No. 22178195 T.W., 22005170 X.C., 22275110 X.C.) are gratefully appreciated.

## Author contributions

Conceptualization: B.L., X.C., T.W. Methodology: B.L., X.C., N.H., S.L. Investigation: B.L., X.C. Visualization: B.L., X.C. Supervision: X.C., F.W., T.W. Writing—original draft: B.L. Writing—review and editing: B.L., X.C., N.H., S.L., Y.W., X.L., F.W., T.W.

## Competing interests

The authors declare no competing interests.
