## [Peer Review File · Nature Communications]

REVIEWER COMMENTS

Reviewer #1 (Remarks to the Author):

The manuscript entitled "Imaging the Dynamic Influence of Functional Groups on Metal Organic Frameworks" by Tiefeng Wang et al. presents a novel approach that combines iDPC-STEM experimental and first-principles techniques to investigate the rotation properties of various functional groups in the linkers of UiO-66-X. The study also examines the impact of these functional groups on the local rigidity of the framework with respect to CO₂ capture.

One notable finding is the presence of functional benzene rings in the UiO-66 channels with distinct rotation properties, suggesting that the structural flexibility of the framework can be modulated by controlling the orientation of these functional groups. Furthermore, the manuscript highlights the impressive relationship between flexibility and local rigidity with CO₂ capture, which has important implications for optimizing the efficiency of CO₂ adsorption in metal organic frameworks.

However, there are some questions should be addressed prior to further evaluation

1) To promote consistency and clarity throughout the manuscript, it would be beneficial to include color descriptions for each icon in the atomic models presented in the figures. Specifically, it is recommended to follow the example set in Figure 4, where the color-coded icons are described in the legend.

2) In regard to Figure 2f and Figure 3e, it appears that the intensity of the center benzene ring in the simulated iDPC-STEM image appears slightly weaker than the four surrounding rings. This difference in intensity may be unexpected, it may be worthwhile for the authors to investigate this issue further and ensure that the simulated images accurately reflect the expected intensity of the benzene rings.

3) In Figure 3 and Figure S7, it appears that the iDPC-STEM images of UiO-66-H have a relatively poorer quality compared to the images of UiO-66-Br and UiO-66-CH₃. Despite this difference in quality, the standard deviation values for these images appear to be similar. If the full width at half maximum (FWHM) values are relative to the rotation angle, it would be expected that the poorer image quality of UiO-66-H would result in a higher standard deviation compared to the other images displayed. To address this issue, it may be beneficial to apply image filters or to replace the images with higher quality versions.

4) In addition to discussing the full width at half maximum (FWHM) of the intensity profile, it would also be beneficial to examine the peaks of the line profile in more detail.

Reviewer #2 (Remarks to the Author):

This manuscript tries to show the aspect of dynamic influence, especially in the direct imaging of dynamic changes after functionalization of MOFs, in order to see correlation between local rigidity and the functional groups on the organic linkers. The (iDPC-STEM) images are beautiful and well-combined with the measurement of FWHM in a rigorous manner and supported by a DFT study. However, this study is focused on pristine materials without the interactions of the MOFs with CO₂ molecules. There are several works where the flexibility of the organic linker has been observed with in-situ experiments. Moreover, the interactions of MOFs with molecules (including CO₂) can promote rotation itself in the organic linker during the sorption process (from pristine materials to material with CO₂ sorbed) (see J. Am. Chem. Soc. 2013, 135, 15986–15989). Besides, authors correlate the MOFs' rigidity to the macroscopic properties of CO₂ uptake. In addition, authors indicate that MOFs functionalization can change the dynamic rotation properties. However, CO₂ has associated a quadrupolar moment that is reflected in a high interaction with polarizable atoms such as O and N. In fact, MOFs based on N and O (UiO-66-NH₂ and UiO-66-OH, respectively) provide the best values for CO₂ uptake. In this sense, the proposed relationship of rotation energy and CO₂ uptake does not fix suitable (see Figure 5c), this is not well supported with these data.

Authors need to carry out an in-situ study of CO₂ sorption that reinforces this hypothesis. For instances, inelastic- and quasi-elastic neutron scattering is one of the most accurate techniques to study it (Chem. Sci., 2017, 8, 3109–3120). This paper is not suitable for Nature Communications.

Reviewer #3 (Remarks to the Author):

This manuscript used iDPC-STEM to characterize the dynamic properties of MOFs, which is novel and interesting. The authors analyzed the n-flipping properties of UiO-66 with different functional groups and systematically studied the influences on the MOF flexibility. The local rigidity of MOFs are also correlated with the CO₂ capacity. This work is appealing and inspiring. Therefore, I suggest that this manuscript can be published in Nature Communications with a minor revision. The following are detailed suggestions.

1. The authors used TEM simulations to correlate with the experimental images. The simulation details were described in the main text, but it is not intuitive. Some supporting figures should be added to better illustrate the simulated models.
2. The authors fitted the FWHM against the calculated rotation energy in Fig. 5. Although the values of fitted parameters were reported, the error range is not included. I recommend the authors to add this into SI.
3. The authors mentioned that NMR can be used to characterize the benzene rotation of BDC linkers. And in this work, the authors proposed that iDPC-STEM can also characterize the flexibility of MOFs. What is the difference between these two methods? And what unique advantages does the iDPC technology have?
4. The authors calculated the intramolecular hydrogen bonds in BDC-CH₃, BDC-NH₂ and BDC-OH. The model structure of UiO-66-OH is shown in Fig. 4, while the other two structures are not included. Are there any structure differences?
5. The authors should add more calculation details of COHP.

RESPONSE TO REVIEWERS' COMMENTS

Reviewer #1

The manuscript entitled “Imaging the Dynamic Influence of Functional Groups on Metal Organic Frameworks” by Tiefeng Wang et al. presents a novel approach that combines iDPC-STEM experimental and first-principles techniques to investigate the rotation properties of various functional groups in the linkers of UiO-66-X. The study also examines the impact of these functional groups on the local rigidity of the framework with respect to CO₂ capture.

One notable finding is the presence of functional benzene rings in the UiO-66 channels with distinct rotation properties, suggesting that the structural flexibility of the framework can be modulated by controlling the orientation of these functional groups. Furthermore, the manuscript highlights the impressive relationship between flexibility and local rigidity with CO₂ capture, which has important implications for optimizing the efficiency of CO₂ adsorption in metal organic frameworks.

However, there are some questions should be addressed prior to further evaluation

1) To promote consistency and clarity throughout the manuscript, it would be beneficial to include color descriptions for each icon in the atomic models presented in the figures. Specifically, it is recommended to follow the example set in Figure 4, where the color-coded icons are described in the legend.

Authors Reply: We thank the reviewer for the constructive suggestions. We added the descriptions in figure legends.

“Figure 1. Crystal structure of UiO-66-X. (a) Schematic model of UiO-66. The views from [110] and [001] projections are shown below. The long and short axes from the [110] projection are marked by blue and green arrows, respectively. The grey, red and white atoms represent C, O and H elements, respectively. (b) XRD patterns of UiO-66-X and simulated results of UiO-66-H.” (Page 15)

“Figure 2. Characterizations of UiO-66-NH₂ and UiO-66-Br. (a, d) HAADF-STEM images and corresponding EDS mapping of UiO-66-NH₂ (a) and UiO-66-Br (d). (b, e) iDPC-STEM images of UiO-66-NH₂ (b) and UiO-66-Br (e) from [110] orientation. The FFT pattern is shown as inset. (c, f) Magnified iDPC-STEM images and simulated results of UiO-66-NH₂ (c) and UiO-66-Br (f). The intensity profile along the long axis and the model of BDC-NH₂ rotation are shown below. The dark lines in the intensity profile are experimental results, while the light lines are the simulated results. The scale bar is 1 nm. The grey, red, white, blue and brown atoms represent C, O, H, N and Br

elements, respectively.” (Page 16)

2) In regard to Figure 2f and Figure 3e, it appears that the intensity of the center benzene ring in the simulated iDPC-STEM image appears slightly weaker than the four surrounding rings. This difference in intensity may be unexpected, it may be worthwhile for the authors to investigate this issue further and ensure that the simulated images accurately reflect the expected intensity of the benzene rings.

Authors Reply: The weaker intensity is attributed to the signal broadening caused by the rotation of benzene, and the stronger intensity of surrounding linkers is observed because we did not change the rotation orientations of these linkers during the simulation. For the simulation, we build a three-layer model of UiO-66-X. The average rotation degrees of different BDC-X linkers are calculated using $\frac{0.18 \text{ eV}}{E_{\text{Rotate}}} \times \frac{\pi}{2}$, as described in Figure 5. For example, the value for BDC-Br is 75°. Considering that the BDC linkers have a distribution of different rotation degrees during observation, we rotate the BDC-Br linkers in the first and third layers by -75° and 75°, respectively, while the other linkers remain unchanged. Therefore, we observe a signal broadening effect at the center benzene ring and thus a weaker intensity. The surrounding rings, however, show stronger intensity because the corresponding BDC linkers are not rotated during the simulation. This is because the surrounded four BDC linkers are not parallel to the [110] direction. Without rotation, they have a 30° angle to the image, making the signal changes after rotation too complex and difficult to analyze.

For clarification, we added the details of simulation as Fig. S9 in SI.

Fig. S9. Typical model of UiO-66-Br for iDPC-STEM simulations. (A) Top view of the three-layered model for simulation. The displayed region in Figures 2 and 3 is marked by the red square. The region highlighted by blue is attributed to the rotated BDC-Br linkers. The BDC-Br linkers in other cells remain unchanged. (B) Side view of rotated BDC-Br linkers. For simulation, we rotate the linker in the first slab by -75° and the linker in the third slab by 75° , while the BDC-Br linker in the second slab remains unchanged. The rotation degree of $\pm 75^\circ$ is calculated using 0.18 eV as the thermal motion energy. We use this model to simulate the distribution of different BDC-Br rotation angles. The grey, red, white and brown atoms represent C, O, H and Br elements, respectively. (SI Page S10)

3) In Figure 3 and Figure S7, it appears that the iDPC-STEM images of UiO-66-H have a relatively poorer quality compared to the images of UiO-66-Br and UiO-66-CH₃. Despite this difference in quality, the standard deviation values for these images appear to be similar. If the full width at half maximum (FWHM) values are relative to the rotation angle, it would be expected that the poorer image quality of UiO-66-H would result in a higher standard deviation compared to the other images displayed. To address this issue, it may be beneficial to apply image filters or to replace the images with higher quality versions.

Authors Reply: We thank the reviewer for this valuable comment. The absolute values of standard deviation of UiO-66-H, UiO-66-CH₃ and UiO-66-Br are close, which are 0.70, 0.61 and 0.64 Å, respectively. However, the difference in relative deviations is more obvious, because of the different average FWHM. The relative bias for UiO-66-H, UiO-66-CH₃ and UiO-66-Br are 24.2%, 19.2% and 17.3%, respectively. As expected, the poorer image quality of UiO-66-H did result in a larger deviation. According to the reviewer's suggestion, we applied image filters to increase the quality of UiO-66-H (**Fig. 3a**), and the revised figure is shown below. The results of average FWHM and corresponding discussion remain the same.

Figure R1. The modification of the iDPC-STEM image of UiO-66-H in Fig. 3a.

Figure 3. Rotation properties of benzene ring in UiO-66-X. (a to e) iDPC-STEM images, simulated

results and intensity profile along the long axis of UiO-66-H (a), UiO-66-CH₃ (b), UiO-66-OH (c), UiO-66-F (d) and UiO-66-Cl (e). The dark lines in the intensity profile are experimental results, while the light lines are the simulated results. (f) Summarized experimental and simulated FWHM of benzene rings along the long axis in UiO-66-X. The experimental FWHM is the averaged value obtained from more than 20 cells, and the error bar is the standard deviation.

4) In addition to discussing the full width at half maximum (FWHM) of the intensity profile, it would also be beneficial to examine the peaks of the line profile in more detail.

Authors Reply: We have analyzed the intensity profile in our previous works and identified the position of carbon atoms in BDC-X linkers (*Nano Lett.* 2023, 23, 1787–1793). Other works have also used iDPC-STEM to reveal the local distortion of UiO-66 (*Nano Lett.* 2022, 24, 9928–9934). In this work, we focused on the rotation properties of benzene ring in BDC-X linkers. Therefore, we mainly studied the FWHM of the intensity profile. In the revised manuscript, we added some discussion on the peak position and the peak shape in the line profiles.

“For UiO-66-Br, however, the signals of BDC-Br are blurred from a narrow line to a larger bright spot (Fig. 2f). The asymmetric peak shape indicates that the functional groups of -Br may be biased in one direction. Moreover, the FWHM along the long axis increased to 4.02 Å. The broadening effect is attributed to the π -flipping of benzene rings in the BDC-Br linkers.” (Page 7)

Reviewer #2

This manuscript tries to show the aspect of dynamic influence, especially in the direct imaging of dynamic changes after functionalization of MOFs, in order to see correlation between local rigidity and the functional groups on the organic linkers. The (iDPC-STEM) images are beautiful and well-combined with the measurement of FWHM in a rigorous manner and supported by a DFT study. However, this study is focused on pristine materials without the interactions of the MOFs with CO₂ molecules. There are several works where the flexibility of the organic linker has been observed with in-situ experiments. Moreover, the interactions of MOFs with molecules (including CO₂) can promote rotation itself in the organic linker during the sorption process (from pristine materials to material with CO₂ sorbed) (see *J. Am. Chem. Soc.* 2013, 135, 15986–15989). Besides, authors correlate the MOFs' rigidity to the macroscopic properties of CO₂ uptake. In addition, authors indicate that MOFs functionalization can change the dynamic rotation properties. However, CO₂ has associated a quadrupolar moment that is reflected in a high interaction with polarizable atoms such as O and N. In fact, MOFs based on N and O (UiO-66-NH₂ and UiO-66-OH, respectively) provide the best values for CO₂ uptake. In this sense, the proposed relationship of rotation energy and CO₂ uptake does not fit suitable (see Figure 5c), this is not well supported with these data. Authors need to carry out an in-situ study of CO₂ sorption that reinforces this hypothesis. For instances, inelastic- and quasi-elastic neutron scattering is one of the most accurate techniques to study it (*Chem. Sci.*, 2017, 8, 3109–3120). This paper is not suitable for Nature Communications.

Authors Reply: We thank the reviewers for their recognition of our images and studies, as well as for their valuable suggestions. Our work focused more on the use of iDPC-STEM to image the dynamic properties of UiO-66 with different functional groups. At the end of our work, we found a relationship between local rigidity and CO₂ uptake. It can partially explain the different CO₂ capture capability of UiO-66-X, but the detailed mechanism requires further investigation.

The reviewer mentioned that CO₂ could possibly interacted with more polarizable atoms, such as O and N. Actually, Schwerdtfeger et al. (*Mol. Phys.* 2019, 117, 1200-1225) reported the polarizabilities of all elements. Using these data, we plotted the CO₂ uptake against the atomic polarizabilities, as shown in **Figure S12**. No obvious correlation was found, indicating that CO₂ capture is not decisively affected by the polarizabilities of functional atoms. For example, along the periodic table of elements, the polarizability of elements increases following the trend of C (11.3) > N (7.4) > O (5.3) > F (3.74), but the CO₂ capture does not follow the same trend, indicating that some other factors also influence the CO₂ capture.

Finally, the recommendations are inspiring and we added the two references in the revised manuscript. We agree that more investigation is required to further reveal the mechanism between

rigidity and CO₂ capture of UiO-66-X. *In situ* inelastic- and quasi-elastic neutron scattering is one of the most accurate techniques for a better understanding of the mechanism, which could provide detailed information of the lattice dynamics and molecular vibrations.

To address the issues raised by the reviewer, we made the following revisions to the text and added **Fig. S12** in SI.

Fig. S12. Correlation between CO₂ uptake and polarizabilities of functional atoms. The polarizabilities are obtained from Ref.⁵⁰ and the values of CO₂ uptake are obtained from Ref.¹³ (SI Page S13)

“Inspired by the fact that CO₂ adsorption can promote the flexibility of MOFs⁴⁹. In this work, here we will briefly discuss the potential impacts of UiO-66-X on CO₂ capture, whose capability is highly related to their local rigidity.” (Page 12)

“Our results agree with this view, and the most rigid UiO-66-OH sample has the highest CO₂ uptake amount. Although CO₂ is considered as acidic and has a quadrupole moment, instead of the most basic UiO-66-NH₂ or the most polarizable UiO-66-Br did not show the highest CO₂ adsorption⁵⁰ (Fig. S11). Instead, the more rigid of the UiO-66-X sample is, the more CO₂ it can adsorb generally. T” (Page 13)

“Moreover, when doping BDC-NH₂ to UiO-66-H, a higher fraction of BDC-NH₂ linkers leads to larger CO₂ uptake⁵², which is also consistent with the trend that higher rigidity tend to benefit the adsorption of guest molecules. However, further investigation is required to better understand the mechanism of how local rigidity influences the CO₂ capture, and *in situ* inelastic- and quasi-elastic neutron scattering is one of the most accurate techniques⁵³.” (Page 13)

Reviewer #3

This manuscript used iDPC-STEM to characterize the dynamic properties of MOFs, which is novel and interesting. The authors analyzed the π -flipping properties of UiO-66 with different functional groups and systematically studied the influences on the MOF flexibility. The local rigidity of MOFs are also correlated with the CO₂ capacity. This work is appealing and inspiring. Therefore, I suggest that this manuscript can be published in Nature Communications with a minor revision. The following are detailed suggestions.

1. The authors used TEM simulations to correlate with the experimental images. The simulation details were described in the main text, but it is not intuitive. Some supporting figures should be added to better illustrate the simulated models.

Authors Reply: We thank the review for valuable suggestions. We added the detailed description of the simulation in SI, and provided a model figure for the simulated regions.

Fig. S9. Typical model of UiO-66-Br for iDPC-STEM simulations. (A) Top view of the three-layered model for simulation. The displayed region in Figures 2 and 3 is marked by the red square. The region highlighted by blue is attributed to the rotated BDC-Br linkers. The BDC-Br linkers in

other cells remain unchanged. (B) Side view of rotated BDC-Br linkers. For simulation, we rotate the linker in the first slab by -75° and the linker in the third slab by 75° , while the BDC-Br linker in the second slab remains unchanged. The rotation degree of $\pm 75^\circ$ is calculated using 0.18 eV as the thermal motion energy. We use this model to simulate the distribution of different BDC-Br rotation angles. The grey, red, white and brown atoms represent C, O, H and Br elements, respectively. (SI Page S10)

2. The authors fitted the FWHM against the calculated rotation energy in Fig. 5. Although the values of fitted parameters were reported, the error range is not included. I recommend the authors to add this into SI.

Authors Reply: We added the error range of fitted parameters in SI.

Table S3. Fitted results of FWHM against rotation energies. (Page S16)

Parameters	$d/\text{\AA}$	E_0/eV	$d_{atom}/\text{\AA}$
Fitted results	1.89 ± 0.22	0.18 ± 0.06	1.82 ± 0.26

“The fitting parameters are d , E_0 and d_{atom} (Table S3).” (Page 11)

3. The authors mentioned that NMR can be used to characterize the benzene rotation of BDC linkers. And in this work, the authors proposed that iDPC-STEM can also characterize the flexibility of MOFs. What is the difference between these two methods? And what unique advantages does the iDPC technology have?

Authors Reply: The NMR method provides only indirect and statistical information of the MOF samples, while the iDPC-STEM can image the rotation properties of BDC-X linkers in real space and provide direct and local information. Our work presents the first visualization of the rotation properties of benzene ring in UiO-66 type MOFs. It also represents pioneering characterization of the dynamic properties of MOFs using electron microscopy. For the characterization of encapsulated materials with different rigidity, such as UiO-66-NH₂@UiO-66-Br, the iDPC-STEM can provide more local information, which can hardly be obtained using NMR.

4. The authors calculated the intramolecular hydrogen bonds in BDC-CH₃, BDC-NH₂ and BDC-OH. The model structure of UiO-66-OH is shown in Fig. 4, while the other two structures are not included. Are there any structure differences?

Authors Reply: After structural optimization, we found that the structure of UiO-66-NH₂ and UiO-66-CH₃ remained unchanged (Fig. S10). The BDC-X linkers are basically the same, while the intramolecular hydrogen bonds had significantly different length. The different strength of hydrogen bonds affected the rotation properties, which was discussed in the ‘theoretical explanations’ section

in Page 9. For clarification, we added a supplementary figure (Fig. S10) to better illustrate the structure of BDC-CH₃ and BDC-NH₂.

“This is attributed to the intramolecular hydrogen bond of BDC-NH₂ and BDC-OH (Fig. S10). The distance between hydroxyl H and carboxyl O in BDC-OH is 1.64 Å, and the bond length between amino H and carboxyl O in BDC-NH₂ is 1.88 Å (Table S2).” (Page 9)

Fig. S10. Schematic diagram of intramolecular hydrogen bond in UiO-66-X. The intramolecular hydrogen bonds are highlighted in red and the H...O atoms are connected by dashed blue lines. The grey, red, white and blue atoms represent C, O, H and N elements, respectively. (SI Page 11)

5. The authors should add more calculation details of COHP.

Authors Reply: We added the calculation details of COHP in methods. Moreover, we added the schematic diagram of the calculated atom pairs in Fig. S11.

“COHP analysis^{46,58} is conducted using LOBSTER to ~~for~~ better understanding the bonding effects between H in functional groups and O in carboxyl groups. The calculated energy range is between -10 and 6 eV.” (Page 21)

Fig. S11. Electronic structure of hydrogen bond. The density of states (DOS) are shown below.

The DOS of adsorbed H is multiplied by 5 times. The overlap of DOS confirmed the existence of hydrogen bonds between O in carboxyl groups (-COO) and H in functional groups (-CH₃, -NH₂, -OH). The crystal orbital Hamilton population (COHP) and the integrated COHP (ICOHP) represent the bonding effects between O in carboxyl groups and H in functional groups. The calculated atom pairs are highlighted in the inset. The positive value of -COHP represent the bonding effect, while the negative value of -COHP represent the anti-boding effect. (SI Page S12)

REVIEWERS' COMMENTS

Reviewer #1 (Remarks to the Author):

The authors have addressed all my concerns. I think the paper is now suitable for publication.

Reviewer #2 (Remarks to the Author):

This manuscript tries to show the aspect of dynamic influence, especially in the direct imaging of dynamic changes after functionalization of MOFs, in order to see correlation between local rigidity and the functional groups on the organic linkers. The (iDPC-STEM) images are beautiful and well-combined with the measurement of FWHM in a rigorous manner and supported by a DFT study. This study has been improved during the revision step. In my opinion, this manuscript is currently suitable for publication in Nature Communications. The authors have made a great effort to address the issues raised by the all referees. I am happy to recommend the publication of the manuscript in its current form.

Reviewer #3 (Remarks to the Author):

The authors have adequately addressed the concerns raised and the quality of the manuscript have been improved after the revision. It could be accepted in its current form.

RESPONSE TO REVIEWERS' COMMENTS

Reviewer #1

The manuscript entitled “Imaging the Dynamic Influence of Functional Groups on Metal Organic Frameworks” by Tiefeng Wang et al. presents a novel approach that combines iDPC-STEM experimental and first-principles techniques to investigate the rotation properties of various functional groups in the linkers of UiO-66-X. The study also examines the impact of these functional groups on the local rigidity of the framework with respect to CO₂ capture.

One notable finding is the presence of functional benzene rings in the UiO-66 channels with distinct rotation properties, suggesting that the structural flexibility of the framework can be modulated by controlling the orientation of these functional groups. Furthermore, the manuscript highlights the impressive relationship between flexibility and local rigidity with CO₂ capture, which has important implications for optimizing the efficiency of CO₂ adsorption in metal organic frameworks.

However, there are some questions should be addressed prior to further evaluation

1) To promote consistency and clarity throughout the manuscript, it would be beneficial to include color descriptions for each icon in the atomic models presented in the figures. Specifically, it is recommended to follow the example set in Figure 4, where the color-coded icons are described in the legend.

Authors Reply: We thank the reviewer for the constructive suggestions. We added the descriptions in figure legends.

“Figure 1. Crystal structure of UiO-66-X. (a) Schematic model of UiO-66. The views from [110] and [001] projections are shown below. The long and short axes from the [110] projection are marked by blue and green arrows, respectively. The grey, red and white atoms represent C, O and H elements, respectively. (b) XRD patterns of UiO-66-X and simulated results of UiO-66-H.” (Page 15)

“Figure 2. Characterizations of UiO-66-NH₂ and UiO-66-Br. (a, d) HAADF-STEM images and corresponding EDS mapping of UiO-66-NH₂ (a) and UiO-66-Br (d). (b, e) iDPC-STEM images of UiO-66-NH₂ (b) and UiO-66-Br (e) from [110] orientation. The FFT pattern is shown as inset. (c, f) Magnified iDPC-STEM images and simulated results of UiO-66-NH₂ (c) and UiO-66-Br (f). The intensity profile along the long axis and the model of BDC-NH₂ rotation are shown below. The dark lines in the intensity profile are experimental results, while the light lines are the simulated results. The scale bar is 1 nm. The grey, red, white, blue and brown atoms represent C, O, H, N and Br

elements, respectively.” (Page 16)

2) In regard to Figure 2f and Figure 3e, it appears that the intensity of the center benzene ring in the simulated iDPC-STEM image appears slightly weaker than the four surrounding rings. This difference in intensity may be unexpected, it may be worthwhile for the authors to investigate this issue further and ensure that the simulated images accurately reflect the expected intensity of the benzene rings.

Authors Reply: The weaker intensity is attributed to the signal broadening caused by the rotation of benzene, and the stronger intensity of surrounding linkers is observed because we did not change the rotation orientations of these linkers during the simulation. For the simulation, we build a three-layer model of UiO-66-X. The average rotation degrees of different BDC-X linkers are calculated using $\frac{0.18 \text{ eV}}{E_{\text{Rotate}}} \times \frac{\pi}{2}$, as described in Figure 5. For example, the value for BDC-Br is 75°. Considering that the BDC linkers have a distribution of different rotation degrees during observation, we rotate the BDC-Br linkers in the first and third layers by -75° and 75°, respectively, while the other linkers remain unchanged. Therefore, we observe a signal broadening effect at the center benzene ring and thus a weaker intensity. The surrounding rings, however, show stronger intensity because the corresponding BDC linkers are not rotated during the simulation. This is because the surrounded four BDC linkers are not parallel to the [110] direction. Without rotation, they have a 30° angle to the image, making the signal changes after rotation too complex and difficult to analyze.

For clarification, we added the details of simulation as Fig. S9 in SI.

Fig. S9. Typical model of UiO-66-Br for iDPC-STEM simulations. (A) Top view of the three-layered model for simulation. The displayed region in Figures 2 and 3 is marked by the red square. The region highlighted by blue is attributed to the rotated BDC-Br linkers. The BDC-Br linkers in other cells remain unchanged. (B) Side view of rotated BDC-Br linkers. For simulation, we rotate the linker in the first slab by -75° and the linker in the third slab by 75° , while the BDC-Br linker in the second slab remains unchanged. The rotation degree of $\pm 75^\circ$ is calculated using 0.18 eV as the thermal motion energy. We use this model to simulate the distribution of different BDC-Br rotation angles. The grey, red, white and brown atoms represent C, O, H and Br elements, respectively. (SI Page S10)

3) In Figure 3 and Figure S7, it appears that the iDPC-STEM images of UiO-66-H have a relatively poorer quality compared to the images of UiO-66-Br and UiO-66-CH₃. Despite this difference in quality, the standard deviation values for these images appear to be similar. If the full width at half maximum (FWHM) values are relative to the rotation angle, it would be expected that the poorer image quality of UiO-66-H would result in a higher standard deviation compared to the other images displayed. To address this issue, it may be beneficial to apply image filters or to replace the images with higher quality versions.

Authors Reply: We thank the reviewer for this valuable comment. The absolute values of standard deviation of UiO-66-H, UiO-66-CH₃ and UiO-66-Br are close, which are 0.70, 0.61 and 0.64 Å, respectively. However, the difference in relative deviations is more obvious, because of the different average FWHM. The relative bias for UiO-66-H, UiO-66-CH₃ and UiO-66-Br are 24.2%, 19.2% and 17.3%, respectively. As expected, the poorer image quality of UiO-66-H did result in a larger deviation. According to the reviewer's suggestion, we applied image filters to increase the quality of UiO-66-H (**Fig. 3a**), and the revised figure is shown below. The results of average FWHM and corresponding discussion remain the same.

Figure R1. The modification of the iDPC-STEM image of UiO-66-H in Fig. 3a.

Figure 3. Rotation properties of benzene ring in UiO-66-X. (a to e) iDPC-STEM images, simulated

results and intensity profile along the long axis of UiO-66-H (a), UiO-66-CH₃ (b), UiO-66-OH (c), UiO-66-F (d) and UiO-66-Cl (e). The dark lines in the intensity profile are experimental results, while the light lines are the simulated results. (f) Summarized experimental and simulated FWHM of benzene rings along the long axis in UiO-66-X. The experimental FWHM is the averaged value obtained from more than 20 cells, and the error bar is the standard deviation.

4) In addition to discussing the full width at half maximum (FWHM) of the intensity profile, it would also be beneficial to examine the peaks of the line profile in more detail.

Authors Reply: We have analyzed the intensity profile in our previous works and identified the position of carbon atoms in BDC-X linkers (*Nano Lett.* 2023, 23, 1787–1793). Other works have also used iDPC-STEM to reveal the local distortion of UiO-66 (*Nano Lett.* 2022, 24, 9928–9934). In this work, we focused on the rotation properties of benzene ring in BDC-X linkers. Therefore, we mainly studied the FWHM of the intensity profile. In the revised manuscript, we added some discussion on the peak position and the peak shape in the line profiles.

“For UiO-66-Br, however, the signals of BDC-Br are blurred from a narrow line to a larger bright spot (Fig. 2f). The asymmetric peak shape indicates that the functional groups of -Br may be biased in one direction. Moreover, the FWHM along the long axis increased to 4.02 Å. The broadening effect is attributed to the π -flipping of benzene rings in the BDC-Br linkers.” (Page 7)

Reviewer #2

This manuscript tries to show the aspect of dynamic influence, especially in the direct imaging of dynamic changes after functionalization of MOFs, in order to see correlation between local rigidity and the functional groups on the organic linkers. The (iDPC-STEM) images are beautiful and well-combined with the measurement of FWHM in a rigorous manner and supported by a DFT study. However, this study is focused on pristine materials without the interactions of the MOFs with CO₂ molecules. There are several works where the flexibility of the organic linker has been observed with in-situ experiments. Moreover, the interactions of MOFs with molecules (including CO₂) can promote rotation itself in the organic linker during the sorption process (from pristine materials to material with CO₂ sorbed) (see *J. Am. Chem. Soc.* 2013, 135, 15986–15989). Besides, authors correlate the MOFs' rigidity to the macroscopic properties of CO₂ uptake. In addition, authors indicate that MOFs functionalization can change the dynamic rotation properties. However, CO₂ has associated a quadrupolar moment that is reflected in a high interaction with polarizable atoms such as O and N. In fact, MOFs based on N and O (UiO-66-NH₂ and UiO-66-OH, respectively) provide the best values for CO₂ uptake. In this sense, the proposed relationship of rotation energy and CO₂ uptake does not fit suitable (see Figure 5c), this is not well supported with these data. Authors need to carry out an in-situ study of CO₂ sorption that reinforces this hypothesis. For instances, inelastic- and quasi-elastic neutron scattering is one of the most accurate techniques to study it (*Chem. Sci.*, 2017, 8, 3109–3120). This paper is not suitable for Nature Communications.

Authors Reply: We thank the reviewers for their recognition of our images and studies, as well as for their valuable suggestions. Our work focused more on the use of iDPC-STEM to image the dynamic properties of UiO-66 with different functional groups. At the end of our work, we found a relationship between local rigidity and CO₂ uptake. It can partially explain the different CO₂ capture capability of UiO-66-X, but the detailed mechanism requires further investigation.

The reviewer mentioned that CO₂ could possibly interacted with more polarizable atoms, such as O and N. Actually, Schwerdtfeger et al. (*Mol. Phys.* 2019, 117, 1200-1225) reported the polarizabilities of all elements. Using these data, we plotted the CO₂ uptake against the atomic polarizabilities, as shown in **Figure S12**. No obvious correlation was found, indicating that CO₂ capture is not decisively affected by the polarizabilities of functional atoms. For example, along the periodic table of elements, the polarizability of elements increases following the trend of C (11.3) > N (7.4) > O (5.3) > F (3.74), but the CO₂ capture does not follow the same trend, indicating that some other factors also influence the CO₂ capture.

Finally, the recommendations are inspiring and we added the two references in the revised manuscript. We agree that more investigation is required to further reveal the mechanism between

rigidity and CO₂ capture of UiO-66-X. *In situ* inelastic- and quasi-elastic neutron scattering is one of the most accurate techniques for a better understanding of the mechanism, which could provide detailed information of the lattice dynamics and molecular vibrations.

To address the issues raised by the reviewer, we made the following revisions to the text and added **Fig. S12** in SI.

Fig. S12. Correlation between CO₂ uptake and polarizabilities of functional atoms. The polarizabilities are obtained from Ref.⁵⁰ and the values of CO₂ uptake are obtained from Ref.¹³ (SI Page S13)

“Inspired by the fact that CO₂ adsorption can promote the flexibility of MOFs⁴⁹. In this work, here we will briefly discuss the potential impacts of UiO-66-X on CO₂ capture, whose capability is highly related to their local rigidity.” (Page 12)

“Our results agree with this view, and the most rigid UiO-66-OH sample has the highest CO₂ uptake amount. Although CO₂ is considered as acidic and has a quadrupole moment, instead of the most basic UiO-66-NH₂ or the most polarizable UiO-66-Br did not show the highest CO₂ adsorption⁵⁰ (Fig. S11). Instead, the more rigid of the UiO-66-X sample is, the more CO₂ it can adsorb generally. T” (Page 13)

“Moreover, when doping BDC-NH₂ to UiO-66-H, a higher fraction of BDC-NH₂ linkers leads to larger CO₂ uptake⁵², which is also consistent with the trend that higher rigidity tend to benefit the adsorption of guest molecules. However, further investigation is required to better understand the mechanism of how local rigidity influences the CO₂ capture, and *in situ* inelastic- and quasi-elastic neutron scattering is one of the most accurate techniques⁵³.” (Page 13)

Reviewer #3

This manuscript used iDPC-STEM to characterize the dynamic properties of MOFs, which is novel and interesting. The authors analyzed the π -flipping properties of UiO-66 with different functional groups and systematically studied the influences on the MOF flexibility. The local rigidity of MOFs are also correlated with the CO₂ capacity. This work is appealing and inspiring. Therefore, I suggest that this manuscript can be published in Nature Communications with a minor revision. The following are detailed suggestions.

1. The authors used TEM simulations to correlate with the experimental images. The simulation details were described in the main text, but it is not intuitive. Some supporting figures should be added to better illustrate the simulated models.

Authors Reply: We thank the review for valuable suggestions. We added the detailed description of the simulation in SI, and provided a model figure for the simulated regions.

Fig. S9. Typical model of UiO-66-Br for iDPC-STEM simulations. (A) Top view of the three-layered model for simulation. The displayed region in Figures 2 and 3 is marked by the red square. The region highlighted by blue is attributed to the rotated BDC-Br linkers. The BDC-Br linkers in

other cells remain unchanged. (B) Side view of rotated BDC-Br linkers. For simulation, we rotate the linker in the first slab by -75° and the linker in the third slab by 75° , while the BDC-Br linker in the second slab remains unchanged. The rotation degree of $\pm 75^\circ$ is calculated using 0.18 eV as the thermal motion energy. We use this model to simulate the distribution of different BDC-Br rotation angles. The grey, red, white and brown atoms represent C, O, H and Br elements, respectively. (SI Page S10)

2. The authors fitted the FWHM against the calculated rotation energy in Fig. 5. Although the values of fitted parameters were reported, the error range is not included. I recommend the authors to add this into SI.

Authors Reply: We added the error range of fitted parameters in SI.

Table S3. Fitted results of FWHM against rotation energies. (Page S16)

Parameters	$d/\text{\AA}$	E_0/eV	$d_{atom}/\text{\AA}$
Fitted results	1.89 ± 0.22	0.18 ± 0.06	1.82 ± 0.26

“The fitting parameters are d , E_0 and d_{atom} (Table S3).” (Page 11)

3. The authors mentioned that NMR can be used to characterize the benzene rotation of BDC linkers. And in this work, the authors proposed that iDPC-STEM can also characterize the flexibility of MOFs. What is the difference between these two methods? And what unique advantages does the iDPC technology have?

Authors Reply: The NMR method provides only indirect and statistical information of the MOF samples, while the iDPC-STEM can image the rotation properties of BDC-X linkers in real space and provide direct and local information. Our work presents the first visualization of the rotation properties of benzene ring in UiO-66 type MOFs. It also represents pioneering characterization of the dynamic properties of MOFs using electron microscopy. For the characterization of encapsulated materials with different rigidity, such as UiO-66-NH₂@UiO-66-Br, the iDPC-STEM can provide more local information, which can hardly be obtained using NMR.

4. The authors calculated the intramolecular hydrogen bonds in BDC-CH₃, BDC-NH₂ and BDC-OH. The model structure of UiO-66-OH is shown in Fig. 4, while the other two structures are not included. Are there any structure differences?

Authors Reply: After structural optimization, we found that the structure of UiO-66-NH₂ and UiO-66-CH₃ remained unchanged (Fig. S10). The BDC-X linkers are basically the same, while the intramolecular hydrogen bonds had significantly different length. The different strength of hydrogen bonds affected the rotation properties, which was discussed in the ‘theoretical explanations’ section

in Page 9. For clarification, we added a supplementary figure (Fig. S10) to better illustrate the structure of BDC-CH₃ and BDC-NH₂.

“This is attributed to the intramolecular hydrogen bond of BDC-NH₂ and BDC-OH (Fig. S10). The distance between hydroxyl H and carboxyl O in BDC-OH is 1.64 Å, and the bond length between amino H and carboxyl O in BDC-NH₂ is 1.88 Å (Table S2).” (Page 9)

Fig. S10. Schematic diagram of intramolecular hydrogen bond in UiO-66-X. The intramolecular hydrogen bonds are highlighted in red and the H...O atoms are connected by dashed blue lines. The grey, red, white and blue atoms represent C, O, H and N elements, respectively. (SI Page 11)

5. The authors should add more calculation details of COHP.

Authors Reply: We added the calculation details of COHP in methods. Moreover, we added the schematic diagram of the calculated atom pairs in Fig. S11.

“COHP analysis^{46,58} is conducted using LOBSTER to ~~for~~ better understanding the bonding effects between H in functional groups and O in carboxyl groups. The calculated energy range is between -10 and 6 eV.” (Page 21)

Fig. S11. Electronic structure of hydrogen bond. The density of states (DOS) are shown below.

The DOS of adsorbed H is multiplied by 5 times. The overlap of DOS confirmed the existence of hydrogen bonds between O in carboxyl groups (-COO) and H in functional groups (-CH₃, -NH₂, -OH). The crystal orbital Hamilton population (COHP) and the integrated COHP (ICOHP) represent the bonding effects between O in carboxyl groups and H in functional groups. The calculated atom pairs are highlighted in the inset. The positive value of -COHP represent the bonding effect, while the negative value of -COHP represent the anti-boding effect. (SI Page S12)